# Prostate Cancer Cells Are Sensitive to Lysosomotropic Agent Siramesine through Generation Reactive Oxygen Species and in Combination with Tyrosine Kinase Inhibitors

**DOI:** 10.3390/cancers14225478

**Published:** 2022-11-08

**Authors:** Emily A. Garcia, Ilsa Bhatti, Elizabeth S. Henson, Spencer B. Gibson

**Affiliations:** 1Department of Biochemistry and Medical Genetics, University of Manitoba Winnipeg, Winnipeg, MB R3T 2N2, Canada; 2CancerCare Manitoba Research Institute, CancerCare Manitoba, Winnipeg, MB R3E 0V9, Canada; 3Department of Oncology, University of Alberta, Edmonton, AB T6G 2R3, Canada; 4Spencer Gibson, Department of Oncology, University of Alberta, Edmonton, AB T6G 2R3, Canada

**Keywords:** lysosomotrophic drug, siramesine reactive oxygen species, lipid peroxidation, apoptosis

## Abstract

**Simple Summary:**

Advanced prostate cancer is often drug resistant and requires new treatment strategies. Lysosomoptropic agents selectively target lysosomes in cancer cells leading to cell death. We found that the lysosome-targeted drug, siramesine, induced cell death in prostate cancer cells lines through lipid peroxidation and, in combination with the kinase inhibitor lapatinib, increases cell death. This provides a novel strategy to treat aggressive prostate cancer cells.

**Abstract:**

Background: Prostate cancer is the most common cancer affecting men often resulting in aggressive tumors with poor prognosis. Even with new treatment strategies, drug resistance often occurs in advanced prostate cancers. The use of lysosomotropic agents offers a new treatment possibility since they disrupt lysosomal membranes and can trigger a series of events leading to cell death. In addition, combining lysosomotropic agents with targeted inhibitors can induce increased cell death in different cancer types, but prostate cancer cells have not been investigated. Methods: We treated prostate cancer cells with lysosomotropic agents and determine their cytotoxicity, lysosome membrane permeabilization (LMP), reactive oxygen species (ROS) levels, and mitochondrial dysfunction. In addition, we treated cells with lysosomotropic agent in combination with tyrosine kinase inhibitor, lapatinib, and determined cell death, and the role of ROS in this cell death. Results: Herein, we found that siramesine was the most effective lysosomotropic agent at inducing LMP, increasing ROS, and inducing cell death in three different prostate cancer cell lines. Siramesine was also effective at increasing cell death in combination with the tyrosine kinase inhibitor, lapatinib. This increase in cell death was mediated by lysosome membrane permeabilization, an increased in ROS levels, loss of mitochondrial membrane potential and increase in mitochondrial ROS levels. The combination of siramesine and lapatinib induced apoptosis, cleavage of PARP and decreased expression of Bcl-2 family member Mcl-1. Furthermore, lipid peroxidation occurred with siramesine treatment alone or in combination with lapatinib. Treating cells with the lipid peroxidation inhibitor alpha-tocopherol resulted in reduced siramesine induced cell death alone or in combination with lapatinib. The combination of siramesine and lapatinib failed to increase cell death responses in normal prostate epithelial cells. Conclusions: This suggests that lysomotropic agents such as siramesine in combination with tyrosine kinase inhibitors induces cell death mediated by ROS and could be an effective treatment strategy in advanced prostate cancer.

## 1. Introduction

Prostate cancer is the most common male cancer. Despite tremendous advancements made on improving early diagnosis and treatment, resistance to current chemotherapy drugs still occurs [1]. Currently, this is an incurable disease and the only strategy available is the use of chemotherapy and androgen deprivation therapy [2,3,4]. Targeted therapies such as tyrosine kinase inhibitors hold promise but are only under clinical trials and not standard of care for prostate cancer. There is need for better therapeutic strategies to treat prostate cancer. 

Lysosomotropic agents are molecules able to penetrate lysosome membranes and induce lysosome membrane permeabilization (LMP). Many of these molecules were designed for clinical use as antihistamines or antidepressants that have LMP properties [5]. Many of these lysosomotropic agents were capable of penetrating and accumulating within lysosomes and inducing cell death through the accumulation of reactive oxygen species in many cancer cell lines [6]. In contrast, lysosomotropic agents are less effective in non-malignant cell lines [7,8]. In addition, these lysosomotropic agents were effective at inducing cell death in in-vitro models of cancer such as breast, lung, chronic lymphocytic leukemia (CLL) and glioblastoma at relatively low doses [7,9,10]. Of these lysosomotropic agents, siramesine is one of the most potent inducers of lipid reactive oxygen species (ROS) and cell death in many different cancer cells [11]. In CLL, acute myeloid leukemia (AML) and breast cancer cells treatment with low doses of siramesine was shown to induce massive lipid peroxidation and mitochondria dysfunction since lipid peroxidation can decrease the mitochondrial membrane potential, leading to cell death [8,10,12]. It is unknown whether prostate cancer cells are as sensitive to lysosomotropic agents alone or in combination with other anti-cancer drugs. 

Lapatinib is a small molecule targeting these two receptors simultaneously and is currently approved by the Food and Drug Administration (FDA) for the treatment of breast cancer. Given its effectiveness in clinical trials for the treatment of breast cancer, and its mechanism of action targeting HER2 receptors altered in prostate cancer, lapatinib was considered as a new strategy for the treatment of advanced prostate cancer [13,14,15]. However, the single therapy therapeutic agent lapatinib was not effective in clinical trials. In one phase II clinical trial, treatment with lapatinib in patients with hormone-sensitive prostate cancer showed no significant antitumor activity although the drug was well tolerated by patients. In a separate phase II trial, a small population of patients with castration-resistant prostate cancer showed some reduction in bone metastasis (7 out of 29) and PSA levels (1 out of 21). Since lapatinib showed small positive results in some patients, combination treatments are therefore being investigated to test its effectiveness [13]. Lysosome-disrupting agents in combination with tyrosine kinase inhibitors were shown to cause synergistic cell death in cancer types such as breast, lung, CLL and glioblastoma cell lines [7,8,9,12,16]. It is still unknown whether a combination with lysosomotropic agents may lead to triggering events of cell death in advanced prostate cancer cells.

Herein, we demonstrate that the lysosomotropic agent, sirasemine, was effective at inducing cell death in a variety of prostate cancer cell lines mediated by ROS. Moreover, the combination of siramesine and Lapatinib induced an increase in ROS, mitochondrial dysfunction and apoptotic cell death in prostate cancers. 

## 2. Materials and Methods

### 2.1. Cell Culture

The three cell lines used were PC3, DU145 and LNCaP which were purchased from ATCC. Cell lines were maintained in a humidified 5% CO_2_ environment (normoxia = 21% O_2_) at 37 °C. PC3 cells were cultured in DMEM/F-12 (Dulbecco’s Modified Eagle Medium/Nutrient Mixture F-12; Gibco, Life Technologies, Indianapolis, IN, USA), DU145 and LNCaP cells were cultured in RPMI 1640 Medium (Gibco, Life Technologies) and all cell lines were supplemented with 5% (*v*/*v*) fetal bovine serum (FBS; Life Technologies) and 1% Penicillin/Streptomycin (Pen/Strep) (Gibco, Life Technologies). Cells were grown in 100 × 20 mm^2^ tissue culture plates (Sarstedt, Newton, MA, USA, Version Five). Generally, cells were passaged upon reaching ~80% confluency (see below). RWPE prostate endothelial cells were purchased from ATCC and grown in Keratinocyte Serum Free Medium (K-SFM) with bovine pituitary extract (BPE) and human recombinant epidermal growth factor (EGF) added per instructions.

### 2.2. Passaging of Cells

All three cell lines were passaged at a 1:4 ratio after reaching ~80% confluency. To detach strongly adherent PC3 cells from culture plates, 3 mL of Trypsin-Ethylenediaminetetraacetic acid (EDTA) (0.05%) (Gibco) was added and incubated at 37 °C for 5–6 min. For DU145 cells, 2 mL of trypsin-EDTA was added and incubated for 5 min. Trypsin-sensitive LNCaP cells were incubated with 2 mL of trypsin for a maximum of 2 min. To stop the proteolytic reaction, cells were neutralized with media up to 10 mL in total. Cells were centrifuged at 1200 rpm for 5 min and resuspended in 10 mL of fresh growth media and dispensed into new 100 × 20 mm tissue culture plates. Fresh growth media was added every 2–3 days. 

### 2.3. Drug Treatments

Drugs were stored in single use aliquots and were used fresh for each experiment. A 24 h dose–response curve was generated for PC3 cells using lysosomotropic agents (siramesine, desipramine, clemastine, loratadine and desloratadine), tyrosine kinase inhibitors (lapatinib, gefitinib and sorafenib), and chemotherapy drugs (paclitaxel and etoposide). A dose–response curve for siramesine was tested in all three cell lines. In addition, the LC_50_ for siramesine, desipramine and clemastine was determined for assays to study their effects on cell lines. For drug combination experiments, the concentrations used were based on the lowest concentration of each drug that induced between 10–15% cell death at 24 h. 

### 2.4. Cell Death Assays

PC3, DU145 and LNCaP cells were added to 12-well plates at a concentration of 1.0 × 10^5^ cells/mL and allowed to grow for 42 h before treatment. On the day of treatment, fresh media was added before the addition of drugs and incubated at 37 °C for 24 h, unless otherwise indicated. For experiments where inhibitors were used, cells were incubated with inhibitors for 1 h before treatment. Depending on the solvent used to dilute drugs, DMSO or water was added as vehicle control. After treatment, cells were collected, together with the media, and resuspended in phosphate saline buffer (PBS) and stained with 10 μL of 0.04% Trypan blue (Sigma, Oakville, ON, Canada). Trypan blue can enter cells through membrane pores of dying cells while being excluded from live ones. Cells were analyzed within 5 min of trypan blue addition using the Novocyte flow cytometer (Acea Biosciences, San Diego, CA, USA). A total of 20,000 events were collected from each sample and gated using the PER-CP channel and data was analyzed using CellQuest software. To detect early and late apoptotic events, cells were resuspended in 1X Binding Buffer with AnnexinV (BD) and 7AAD (BD) dyes diluted in 1X Binding Buffer. Cells were incubated at 37 °C for 15 min and immediately analyzed by flow cytometry. A total of 20,000 events were collected and gated using FITC and PER-CP channels. Events positive for either FITC or PER-CP were considered apoptotic cells. 

### 2.5. Combination Index Method

PC3 cells (1.5 × 10^4^) were seeded per well of a 96-well flat bottom plate and grown overnight with 100 μL DMEM F12 (5% FBS, 1% penicillin-streptomycin). Cells were treated with DMSO as a negative control, and varying concentrations of lapatinib (3–10 μM), siramesine (5–30 μM) or a combination of 10 μM siramesine and 0.5 μM lapatinib in 100 μL media for 24 h. An amount of 10 μL MTS assay reagent (CellTiter 96^®^ AQueous Non-Radioactive Cell Proliferation Assay (MTS), Promega, Madison, WI, USA) was added to each well after treatment, and the plate was incubated for 4 h at 37 °C in the dark before obtaining readings. Wells containing only media were used to normalize background. Percentage cell death was normalized to negative control and plotted against treatment concentrations, and lines of best fit were drawn to obtain the IC_50_ for siramesine or lapatinib. Combination index was calculated using the formula
$$Combination\;Index=\frac{{\left(D\right)}_{1}}{{\left(D_{x}\right)}_{1}}+\frac{{\left(D\right)}_{2}}{{\left(D_{x}\right)}_{2}}$$
where (*D*)_1_ and (*D*)_2_ are the doses of siramesine and lapatinib in combination required to reach the IC_50_, respectively, and (*D_x_*)_1_ and (*D_x_*)_2_ are the doses of siramesine and lapatinib alone that are required to reach the IC_50_ as single agent treatments, respectively. A combination index of <1 indicates synergy.

### 2.6. Cell Viability Assay

PC3 cells (1.5 × 10^4^) were seeded per well of a 96-well flat bottom plate and grown overnight with 100 μL DMEM F12 (5% FBS, 1% penicillin-streptomycin). Cells were treated with DMSO as a negative control, 0.5 μM lapatinib, 10 μM siramesine, or a combination of siramesine and lapatinib in 100 μL media for 24 h. An amount of 10 μL MTS assay reagent (CellTiter 96^®^ AQueous Non-Radioactive Cell Proliferation Assay (MTS), Promega) was added to each well after treatment, and the plate was incubated for 4 h at 37 °C in the dark before obtaining readings. Wells containing only media were used to normalize background. Percentage cell death was normalized to negative control.

### 2.7. Membrane Permeability Assay

PC3, DU145 and LNCaP cells were seeded in 12-well plates at a concentration of 1.0 × 10^5^ cells/mL and allowed to grow for 42 h before treatment. Depending on the solvent used to dilute drugs, DMSO or water was added as vehicle control. After 4 h treatment the media was removed from cells and Lysotracker Deep Red dye (50 nM, Invitrogen, Carlsbad, CA, USA) was diluted in fresh media and added to cells for 15 min at 37 °C in the dark. Cells were collected, together with the media, and resuspended in PBS and analyzed using the Novocyte flow cytometer (Acea Biosciences). An amount of 20,000 events were collected from each sample and gated using the PER-CP channel. Data was analyzed using CellQuest software.

### 2.8. Lysotracker Assay

PC3 cells (3 × 10^5^) were seeded per 60 mm plate on glass coverslips and treated for 4 h with DMSO as a negative control, 0.5 μM lapatinib, 10 μM siramesine or a combination of siramesine and lapatinib in 2 mL DMEM F12 (5% FBS, 1% penicillin-streptomycin) media. Cells were stained for 30 min at 37 °C in the dark with 100 nM LysoTracker^TM^ Red DND-99 (Invitrogen) in 2 mL media, washed with PBS, and fixed for 20 min with 1 mL paraformaldehyde at room temperature. Cells were mounted with 10 μL mounting media containing DAPI. Images were obtained by confocal microscopy.

### 2.9. ROS Detection Assay

PC3, DU145, and LNCaP cells were seeded in 12-well plates at a concentration of 1.0 × 10^5^ cells/mL and allowed to grow for 42 h before treatment. Depending on the solvent used to dilute drugs, DMSO or water was added as vehicle control. After 4 h or 24 h treatment, cells were collected, together with the media, and resuspended in PBS and stained with 3.2 μM dihydroethidium (DHE, Invitrogen) for 30 min at 37 °C in the dark. Cells were analyzed using the Novocyte flow cytometer (Acea Biosciences) and Cellquest software. An amount of 20,000 events were collected from each sample and gated using the PER-CP channel. 

### 2.10. Mitochondria Membrane Potential Assay

PC3 cells were seeded in 12-well plates at a concentration of 1.0 × 10^5^ cells/mL and allowed to grow for 42 h before treatment. Depending on the solvent used to dilute drugs, DMSO or water was added as vehicle control. After 24 h treatment, cells were collected together with the media and resuspended in PBS and stained with 25 nM tetramethylrhodamine (TMRM, Invitrogen) for 30 min at 37 °C. Cells were analyzed using the Novocyte flow cytometer (Acea Biosciences) and Cellquest software. An amount of 20,000 events were collected from each sample and gated using the PE channel. 

### 2.11. Mitochondria Superoxide Detection Assay

PC3 cells were seeded in 12-well plates at a concentration of 1.0 × 10^5^ cells/mL and allowed to grow for 42 h before treatment. Depending on the solvent used to dilute drugs, DMSO or water was added as vehicle control. After 24 h treatment, cells were collected together with the media and resuspended in PBS and stained with 5 μM Mitosox Red (Invitrogen) for 10 min at 37 °C in the dark. Cells were analyzed using the Novocyte flow cytometer (Acea Biosciences) and Cellquest software (Version five). An amount of 20,000 events were collected from each sample and gated using the PE channel. 

### 2.12. Lipid Peroxidation Assay

PC3 cells were seeded in 12-well plates at a concentration of 1.0 × 10^5^ cells/mL and allowed to grow for 42 h before treatment. Depending on the solvent used to dilute drugs, DMSO or water was added as vehicle control. After 24 h treatment, cells were collected together with the media and resuspended in PBS and stained with 1 μM C11-BODIPY (Invitrogen) for 30 min at 37 °C in the dark. Cells were analyzed using the Novocyte flow cytometer (Acea Biosciences) and Cellquest software. An amount of 20,000 events were collected from each sample and gated using the PE channel.

### 2.13. Western Blot

PC3 cells were cultured as described above and seeded in 6-well plates at a concentration of 3.0 × 10^5^ cells/mL, and allowed to grow for 42 h before treatment. Depending on the solvent used to dilute drugs, DMSO or water was added as vehicle control. After treatment, cells were collected and lysed in 1% NP40 Lysis buffer. Protein was quantified using the Denovix DS-11 UV-Vis spectrophotometer. Samples were loaded into pre-made 4–20% acrylamide Mini-PROTEAN TGX Stain-Free Protein gels (Bio-rad, Montreal, QC, Canada). PageRuler Plus Prestained Protein ladder (Thermo Fisher, Waltham, WA, USA) was loaded in one well to determine the molecular weight of proteins. Antibodies against PARP was purchased from Thermofisher (cat. #MA3-950) and caspase 3 (cat. #06-735) and tubulin (cat. #T5168) were purchased from Cell Signaling Inc., Boston, MA, USA. The antibodies were incubated on membranes (1:1000 ratio) per manufacture instructions. Membranes were developed by enhanced chemiluminescence using the Pierce ECL Western Blotting substrate (Thermo Fisher) according to manufacturer’s instructions. The membranes were imaged using the ImageQuant LAS 500 gel imager (GE Healthcare Life Sciences, Chicago, IL, USA). For all western blot membranes, the reference gene, Actin, was used as a loading control. Protein images obtained from the gel imager were quantified using ImageJ software (v. 2.0). 

### 2.14. Statistical Analysis

All graphs were generated using GraphPad Prism 7. Statistical analysis for flow cytometry experiments were conducted using GraphPad Prism 7. Statistical significance was determined using a two-tailed unpaired *t*-test for all treatments and control samples. A *p*-value < 0.05 was considered statistically significant (represented by *) in addition to a *p*-value < 0.01 (represented by **), and a *p*-value < 0.001 (represented by ***). Error bars represent standard error of the mean for each treatment, and a minimum of three independent replicates were included for each experiment. For normalization of protein expression levels, Microsoft Excel 2017 was used. 

## 3. Results

### 3.1. Lysosomotropic Agents Induce Lysosome Membrane Permeabilization in Prostate Cancer Cells

Lysosomotropic agents are weak bases that accumulate within lysosomes and can induce pores in the membranes leading to the release of their toxic contents into the cytosol [6,17]. Two types of lysosomotropic agents were tested: H1 antihistamines, clemastine, loratadine and desloratadine, and the ASM inhibitors desipramine and siramesine. To determine whether these compounds were able to induce LMP in aggressive and drug resistant prostate cancer cell lines, PC3 cells were treated with increasing doses of these lysosomotropic agents for 4 h and analyzed by flow cytometry using the Lysotracker fluorescent dye. A decrease in fluorescence indicated an increase in LMP. Siramesine was used to treat prostate cells at concentrations of 10, 15 and 20 μM. At these concentrations, siramesine induced statistically significant levels of LMP in a dose-dependent manner up to 12% (Figure 1a). The second most potent LMP inducer was loratadine. Prostate cancer cells were treated with loratadine at concentrations of 10, 25 and 50 μM and showed a significant 11% increase in LMP only at the highest concentration (Figure 1b). LMP induced by desipramine reached 8% only at the highest concentration (Figure 1c). The H1 antihistamine desloratadine failed to significantly induce LMP (Figure 1d). Lastly, treatment of prostate cancer cells with 10, 60 and 80 μM of clemastine did not induce a statistically significant increase in LMP (Figure 1e). From these results, the most potent lysosome disrupting agents were siramesine followed by loratadine and desipramine. 

Since siramesine was the most potent lysosome-disrupting agent in PC3 cells, LMP was tested in the other two prostate cancer cell lines, DU145 and LNCaP. DU145 showed the most significant percentage of LMP when treated with 10, 35 and 45 μM siramesine reaching up to 75% at the highest dose (Figure 1f). On the other hand, when LNCaP cells were treated with 10, 25 and 40 μM siramesine, LMP significantly increased up to 33% (Figure 1g). DU145 and LNCaP cells showed higher levels of LMP than PC3 cells, and this effect was most pronounced in DU145 cells (Figure 1f,g). 

### 3.2. Lysosomotropic Agents Induce ROS in Prostate Cancer Cells

Lysosome membrane permeabilization can lead to significant increases in reactive oxygen species resulting in cell death [12,17,18]. To determine if the most potent LMP inducers could increase significant ROS levels in prostate cancer cells, PC3 were treated for 4 h and stained with DHE, a dye that changes fluorescence when it reacts with superoxide. After treating cells with 10, 15 and 20 μM siramesine, ROS levels significantly increased up to 40% and this effect was similar at 15 and 20 μM (Figure 2a). On the other hand, desipramine-induced non-significant levels of ROS at 10, 100 and 150 μM, and while it increased ROS levels in a dose-dependent manner, it induced 20% less ROS than siramesine even at the highest concentration tested (Figure 2b). When cells were treated with clemastine at 10, 60 and 80 μM concentrations, significant ROS levels were observed only at 80 μM. Clemastine showed the lowest levels of ROS compared to the other two drugs, with increases up to 8% (Figure 2c). From these results, siramesine was the most potent ROS inducer even at the lowest concentration tested. 

To determine the ability of siramesine to induce similar ROS levels in the other prostate cancer cell lines, DU145 and LNCaP cells were treated with increasing concentrations of siramesine for 4 h and stained with DHE, followed by flow cytometry analysis. Siramesine induced the least amount of ROS in DU145 cells compared to PC3 cells with only a 20% significant increase in ROS when treated with 35 and 45 μM concentrations (Figure 2d). In LNCaP cells, siramesine significantly induced the highest increase in ROS when treated with 10, 25 and 40 μM concentrations and this effect was observed in a dose dependent manner, reaching a maximum of 55% increase in fluorescence (Figure 2e). This showed significant increases in ROS levels when treated with siramesine, and LNCaP cells showed the highest increase followed by PC3 and DU145 cells. 

### 3.3. Siramesine Was the Most Potent Lysosomotropic Agent to Induce Cell Death

To determine which lysosomotropic agent induced the highest percentage of cell death in prostate cancer cell lines, a death curve was generated in PC3 cells using the most potent LMP and ROS inducers previously tested. To determine the amount of cell death, cells were treated for 24 h and stained with Trypan blue, a dye that stains cells with damaged plasma membranes, which is an indicator of cell death. Siramesine (10–50 μM) and desipramine (10–100 μM) induced cell death in a dose-dependent manner with an LC_50_ of 20 μM and 100 μM, respectively (Figure 3a,b). The ASM inhibitor siramesine was the most potent drug to induce cell death in PC3 cells; therefore, a dose response for siramesine was generated using the other two prostate cancer cell lines DU145 and LNCaP. In these cell lines, siramesine induced less cell death than in PC3 cells (LC_50_ = 35 μM and 40 μM respectively) (Figure 3c,d). From these results, siramesine was the most effective lysosomotropic agent to induce cell death in prostate cancer cells. 

### 3.4. Siramesine Induced Significant Cell Death with the Tyrosine Kinase Inhibitor Lapatinib

Combining lysosomotropic agents with tyrosine kinase inhibitors has previously been reported to induce synergistic cell death in several cancer types such as breast, lung and glioblastoma [7,8,9]. From the initial results, the ASM inhibitors siramesine and desipramine were selected for combination experiments with tyrosine kinase inhibitors, since these two drugs were the most potent at inducing LMP, ROS and cell death in PC3 cells. The concentration used for combination treatments was selected based on the lowest concentration of each drug that induced between 5–10% cell death. After treating PC3 cells with 10 μM of siramesine and 0.5 μM lapatinib, cell death increased approximately 70% (Figure 4a). This combination appeared to significantly increase cell death since the additive effect of these two drugs alone was approximately 15% compared to 70% when combined together (Figure 4a). We then determine the combination index (CI) where a value of less than one is synergistic cell death. The combination of siramesine and lapatinib gave a CI of 0.7. To confirm the combination of siramesine and lapatinb increased cell death, we determined cell viability with an MTS cell death assay. This showed siramesine and lapatinib increased cell death from 10% in untreated cells to 36% in cells treated in combination (Appendix A). When 10 μM desipramine was combined with 0.5 μM lapatinib, there was no significant increase in cell death observed, since cell death increased 6% with desipramine alone and 5% with lapatinib alone, and the addition of the two drugs combined increased cell death by only 15% (Figure 4b). Since only siramesine increased cell death in PC3 cells in combination with lapatinib, the combination was tested in the other two cell lines at the same concentration. Signiant cell death was also observed in DU145 and LNCaP cells when siramesine and lapatinib were treated but the amount of cell death was lower than PC3 cells (Figure 4c,d). In DU145 cells, cell death increased up to 32% (Figure 4c) and in LNCaP cells increased to 40% (Figure 4d). Based on these results, siramesine but not desipramine in combination with lapatinib increased cell death in all prostate cancer cell lines and this effect was more prominent in the advanced prostate cancer cell line PC3, followed by DU145 and LNCaP cells. 

To determine whether increased in cell death was specific to lapatinib or if other tyrosine kinase inhibitors increase cell death, PC3 cells were treated with 10 μM sorafenib or 10 μM gefitinib. The results showed increased cell death occurred with siramesine in combination with sorafenib but not with gefitinib. Combination with sorafenib showed significant increased cell death since the additive effect of these two drugs alone was approximately 11% compared to 78% when combined (Appendix A). However, when 10 μM siramesine was combined with 10 μM gefitinib, there was no significant increase in cell death, since cell death increased 6% with gefitinib alone and 5% with siramesine alone, and the addition of the two drugs combined increased cell death by only 14% (Appendix A). Lastly, to investigate whether this combination strategy was also capable of increased cell death with topsoisomerase inhibitor etoposide or microtubule stabilizer paclitaxel that have been used to treat metastatic castration-resistant prostate cancer, the drugs [19,20] were tested for combination treatments. A total of 10 μM siramesine in combination with 200 μM etoposide induced 38% cell death and this effect significantly increased cell death since the amount of cell death induced by siramesine alone was 4% and 8% for etoposide (Appendix A). Paclitaxel was not able to induce significant cell death when combined with siramesine, since 1 μM paclitaxel alone induced 11% cell death and 10 μM siramesine induced 3%, but when combined together, cell death decreased to 9% (Appendix A). Thus, lapatinib was the best candidate for combination treatments with tyrosine kinase inhibitors tested.

To determine what type of cell death mechanism is involved after treatment with siramesine and lapatinib, inhibitors for autophagy (3-MA, Spautin-1, BafA1, NH_4_Cl), ferroptosis (Fer-1), necroptosis (Nec-1) and apoptosis (z-VAD) were added 1 h before treatment and cell death was measured at 24 h by Trypan blue and analyzed by flow cytometry. When cells were treated with the autophagy inhibitors 3-MA (2 mM) cell death increased by 10% at 24 h and decreased 6% after 48 h (Appendix A). The necroptosis inhibitor Nec-1 also increased cell death at 24 h by 8% and it did not change the amount of cell death at 48 h (Appendix A). A similar pattern was observed when the ferroptosis inhibitor Fer-1 was added since cell death increased by 10% at 24 h and did not reduce nor increase cell death at 48 h (Appendix A). Lastly, the apoptosis inhibitor z-VAD slightly increased cell death at 24 h by 5% and did not reduce the amount of cell death by siramesine after 48 h (Appendix A). From these results, there was no statistically significant decrease in cell death with any of the inhibitors tested in PC3 cells since the use of inhibitors only resulted in an increase in cell death at 24 h and did not have an effect in cell death after 48 h. There are other more specific ways by which cells can die by apoptosis through a mechanism independent of caspases, which could provide an explanation for the failure of the caspase-dependent inhibitor z-VAD to reduce cell death (Appendix A). To test whether PC3 cells are dying by apoptosis regardless of the role of caspases, cells were treated with 10 μM siramesine and 0.5 μM lapatinib and cell death was analyzed using the apoptotic assay AnnexinV/7AAD. This assay can identify early and late apoptotic events by detecting the apoptotic marker phosphatidyl serine in the outer leaflet of the plasma membrane. After 24 h treatment, there was a significant increase in apoptotic events up to 57% (Figure 5a, Appendix A). To further investigate if these apoptotic events were caspase-dependent or independent, a western blot analysis was performed to measure protein expression levels of PARP and caspase-3. The western blot showed detectable amounts of cleaved PARP and slight reduction in caspase-3 expression following combination treatment (Appendix A). This, however, fails to correlate with the amount of apoptosis (Figure 5a). In addition, western blotting showed significant reduction in the anti-apoptotic protein Mcl-1 after combinational treatment (Appendix A). Taken together, the combination of siramesine and lapatinib induces apoptosis in prostate cancer cells. 

### 3.5. Siramesine in Combination with Lapatinib Induces Lysosome Membrane Permeabilization and ROS Leading to Cell Death

Lysosome membrane permeabilization increased in all prostate cancer cell lines tested when treated with siramesine alone. To further investigate whether LMP is a siramesine-induced event or if this effect increases in combination with lapatinib, LMP was measured at 15 min, 1 h and 4 h. PC3 cells were treated with siramesine alone, lapatinib alone or in combination and incubated with the Lysotracker dye (50 nM) for 30 min and analyzed by flow cytometry. Increases in LMP were time-dependent but these effects showed only a 1% increase after treatment with siramesine alone (Figure 6). When treated with lapatinib, there was only a 1% increase in LMP at 4 h compared to 15 min and 1 h treatments (Figure 6). Lastly, the combination of siramesine and lapatinib induced higher LMP levels than the two (Figure 6). The highest increase in LMP was observed at 4 h where both siramesine and lapatinib induced a 2% LMP and, combined, it increased to 4% (Figure 6). To confirm these results, we stained cells with lysotracker and visualized punctate staining by fluorescent microscopy. We found that 10% cells lost punctate staining of lysosomes whereas 36% of cells treated with siramesine and lapatinib lost punctate staining of lysosomes, indicating lysosome membrane permeabilization (Appendix A). In conclusion, treatment with siramesine and lapatinib showed increases in LMP. 

It has been reported that siramesine increases the levels of ROS and this can be accompanied by increases in lipid peroxidation and mitochondrial damage. After treatment with siramesine and lapatinib for 24 h, there was a significant increase in ROS levels in PC3 cells (Figure 7a). To investigate whether the production of ROS after treatment originates from lipid peroxidation, cells were treated with siramesine and lapatinib and stained for lipid radicals with the dye C11 BODIPY. After 24 h, this combination induced approximately 60% lipid peroxidation (Figure 7b). Siramesine alone also induced a 60% increase in lipid peroxidation but not with lapatinib, which suggests that siramesine alone induces lipid peroxidation (Figure 7b). A significant additive effect was observed when mitochondrial superoxide was measured at 24 h since the combination treatment increased mitochondrial oxide up to 70%, compared to the 12% increase that would be observed if this effect was additive (Figure 7c). Since combination treatment with siramesine and lapatinib increased mitochondrial superoxide levels, the mitochondrial membrane potential was measured after 24 h. A decrease in fluorescence indicates levels of mitochondrial dysfunction. Increased ROS was observed with this combination treatment since super oxide levels after siramesine treatment increased up to 4%, lapatinib levels increased 6% and the combination treatment resulted in a 70% increase in mitochondrial super oxide levels (Figure 7d). Treatment with siramesine and lapatinib suggest significant increases in the levels of reactive oxygen species which include generation of mitochondrial super oxide while decreasing mitochondrial membrane potential. 

To identify whether treatment with siramesine induces lipid peroxidation alone or whether it also targets soluble cellular components, 200 μg/mL of the lipid ROS scavenger alpha-tocopherol was added 1 h prior to treatment with 20 μM siramesine. We found the siramesine alone induced 78% cell death whereas alpha-tocopherol treated cells had 18% cell death. The combination of alpha-tocopherol and siramesine reduced cell death to 48% (Figure 8a). When the amount of ROS was measured after tocopherol treatment, ROS levels decreased from 41% to 17% (Appendix A). When PC3 cells were treated with the combination of siramesine and lapatinib in the presence and absence of alpha-tocopherol, the amount of cell death decreased from 45% to 10% (Figure 8b). This suggests the lipid ROS scavenger alpha-tocopherol was able to reduce siramesine and lapatinib-induced cell death. 

### 3.6. Normal Prostate Epithilial Cells Fail to Give Increase Apoptotic Repsonse following Combination of Siramesine and Lapatinib Treatment

To determine whether normal prostate epithelial cells are resistant to lysosomotropic agents in combination with lapatinib. We treated transformed prostate epithelial cell line RWPE with siramesine, lapatinib and in combination, and determined the amount of total cell death. We found that the combination of siramesine and lapatinib failed to give increased cell death in RWPE cells after 24 h following treatment (Figure 9). 

## 4. Discussion

Drug resistance is the main obstacle for effectively treating advanced prostate cancer. New therapeutic strategies need to be developed. Targeting lysosomes in combination with tyrosine kinase inhibitors to treat aggressive cancer were effective in several in-vitro models of breast, lung, glioblastoma and CLL [7,8,9,10,21]. However, it has not been investigated in prostate cancer cells. We found the lysosomotropic agent, siramesine, was the most effective at inducing cell death in prostate cancer cells through increased lipid peroxidation. When combined with tyrosine kinase inhibitor lapatinib, the amount of cell death significantly increased. This suggests that lysosomotropic agents in combination with tyrosine kinase inhibitors could be an effective treatment strategy in prostate cancer. The limitation of the study is the use of cell lines and whether these treatment doses are clinically achievable. In the future, we will investigate the clinically achievable doses using animal and organoid models. 

The idea of repurposing lysosomotropic agents originally designed to treat health conditions such as depression, or the use of antihistamines with lysosomotropic properties had successful therapeutic results in a wide range of cancer types [22]. Since then, studies have found several more compounds that accumulate within lysosomes that induce cell death in a relatively short time [23]. In CLL cells, treatment with antihistamines (loratadine, desloratadine, clemastine) and antidepressants (siramesine and desipramine) significantly increased cell death compared to non-malignant cells [12,21]. We found clemastine, desipramine and siramesine to induce to cell death in PC3 cells and siramesine was the most effective drug at low concentrations. Antihistamines desloratadine and loratadine did not induce cell death at any of the concentrations tested despite its effectiveness in other cancer types. These differences in results could be due to prostate cancer being more dependent on lysosome function supported by having higher lysosome numbers, size, and the type and level of toxic content inside lysosomes such as labile iron used to provide cancer cells with additional building blocks.

Prostate cancers can be androgen sensitive or resistant. We found that PC3 and DU145 cells that lack the androgen receptor were more sensitive to siramesine, where LNCaP cells that have an androgen receptor were the least sensitive. Androgen receptor has been shown to be a target for TFEB, a master regulator of lysosome biogenesis [24] and might contribute to lysosomotropic agent resistance. This will need to be the focus of future investigation 

Some cancer cell lines possess increased lysosomal genetic alterations in key enzymes such as Hsp70 which provides lysosome membranes with extra protection from lysosome-induced cell death [25]. The lower number of lysosomotropic agents reported to kill prostate cancer cells compared to other cell lines could suggest that either prostate cancer is less susceptible to lysosome damage or antihistamines loratadine and desloratadine were much more effective at killing CLL cells [26]. Perhaps these drugs do not readily diffuse through lysosome membranes and are more readily tolerated and processed efficiently by lysosomes. Siramesine was effective as a lysosomotropic agent in prostate cancer cell lines compared to lysosomotropic agent [7,9]. Siramesine is not FDA-approved for cancer therapy since it was developed as an anti-depressant drug and did not achieve significant clinical results; however, several researchers and this group have found their potential as a lysosomotropic agent and an inducer of cell death in many cancer cells [11,16,27,28]. Breast, CLL, lung and glioblastoma cell lines showed significant cell death after 24 h when treated with the lysosomotropic agent siramesine. In these cases, cell death was associated by an increased in lysosome membrane disruption, lipid ROS and mitochondrial damage [7,8,9,12]. Despite the number of reports conducted on the effects of siramesine on cancer cell lines, its mechanism of action is not fully understood yet as it displays heterogeneity on its effects on cellular processes, which seems to work in a cell- and context-dependent manner [17,23] This will be the focus of future investigations.

Lysosomotropic agents can also induce other cellular events leading to cell death. One of the most widely characterized consequences of lysosome membrane permeabilization is the release of lysosomal proteases into the cytosol, as the membrane becomes porous for lysosome contents to diffuse into the cytosol and among them cathepsin proteases are one of the most toxic enzymes [29]. Their function is to degrade molecules destined to recycling within the lysosome but when they are released into the cytosol, they are capable of activating proteins associated with cell death signals, such as Bid and Bak [30]. Treatment with siramesine triggers the release of cathepsins into the cytosol followed by a decrease in mitochondrial membrane potential and ultimately, cell death [11]. In a study conducted using immortalized keratinocyte HaCaT cells and glioblastoma U-87MG cells, cathepsin release into the cytosol was not observed after treatment with higher concentrations of siramesine than the one used in this study [7,9,12,16]. 

As the accumulation of lysosomotropic agents within lysosomes occurs, this often leads to an increase in ROS as these drugs destabilize lysosome membranes by intercalating within membranes, inhibiting or activating key enzymes, and in some cases working as a detergent [5,31]. Production of ROS is not only tied to lysosomes as it can interfere with cytosolic components or lipid in other organelles, such as mitochondria destabilizing the mitochondria membrane potential [30,32]. We found this to be the case when we treated prostate cancer cell lines with siramesine with significant increases in lipid ROS accompanied by decreased mitochondrial membrane potential and increased levels of mitochondria superoxide. We also observed a decrease in Mcl-1 protein levels that might play a role in mitochondrial dysfunction and will be the focus of future investigation. In the same study conducted with HaCaT and U87 cells, siramesine induced mitochondrial damage and induction of lipid peroxidation. Treatment with the lipophilic antioxidant alpha-tocopherol before addition of siramesine was able to decrease cell death and restore mitochondrial function [16]. In this study, we investigated the role of lipid ROS as a trigger for siramesine-induced cell death by treating with alpha-tocopherol and measuring cell death. Our results agree with the previous study as the same concentration of alpha-tocopherol decreased cell death after 24 h treatment with siramesine. This indicates the main mechanism lysosomotropic agents induce cell death is through increased ROS. The precise mechanism(s) for lysosomotropic agent-induced cell death in prostate cancer cells will be the focus of future studies. 

## 5. Conclusions

Taken together, this provides evidence that lysosomotropic agents can induce cell death through a mechanism dependent on lipid ROS and mitochondrial dysfunction in prostate cancer cells. This could also provide a potential strategy for treating advanced prostate cancer in combination with tyrosine kinase inhibitors such as lapatinib. 

## Figures and Tables

**Figure 1 cancers-14-05478-f001:**
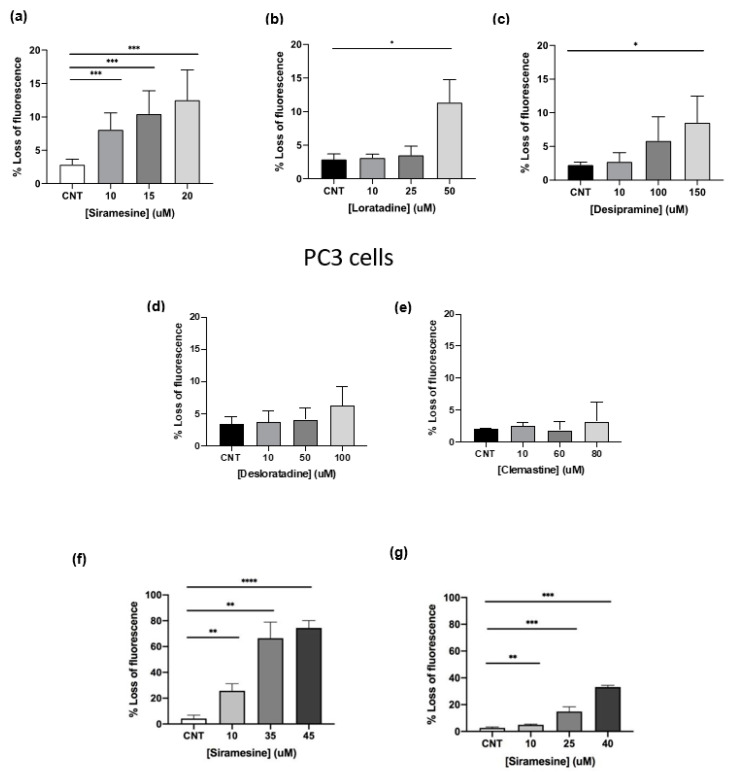
**Lysosomotropic agents induce lysosome membrane permeabilization at 4 h in PC3 cells.** PC3 cells were treated with (**a**) siramesine, (**b**) loratadine, (**c**) desipramine, (**d**) desloratadine and (**e**) clemastine for 4 h and stained with the fluorescent dye Lysotracker (50 nM) and analyzed by flow cytometry. A decrease in fluorescence indicates an increase in LMP. (**f**) DU145 and (**g**) LNCaP cells were treated with increasing doses of siramesine for 4 h and stained with the fluorescent dye Lysotracker (50 nM) and analyzed by flow cytometry. Decrease in fluorescence indicates an increase in LMP. Results are representative of at least three independent replicates (*n* = 3). A *p*-value < 0.05 was considered statistically significant (represented by *) in addition to a *p*-value < 0.01 (represented by **), and a *p*-value < 0.001 (represented by ***).

**Figure 2 cancers-14-05478-f002:**
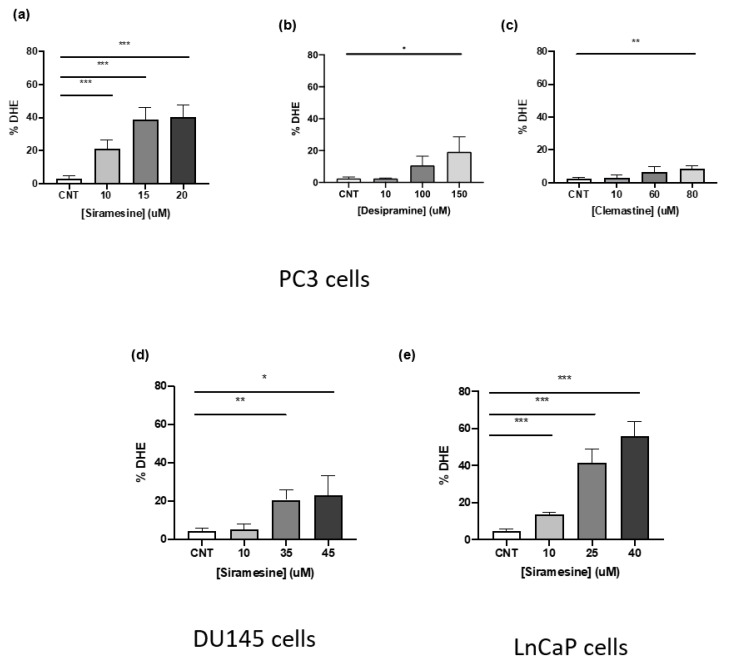
**Siramesine, desipramine and clemastine induce ROS production after 4 h treatment.** PC3 cells were treated with (**a**) siramesine, (**b**) desipramine, (**c**) and clemastine for 4 h, stained with the fluorescent dye dihydroethidium (DHE, 50 nM), and analyzed by flow cytometry. An increase in fluorescence indicates an increase in ROS levels. Results are representative of at least three independent replicates (*n* = 3). (**d**) DU145, and (**e**) LNCaP cells were treated with increasing doses of siramesine for 4 h, stained with the fluorescent dye dihydroethidium (DHE, 50 nM) and analyzed by flow cytometry. An increase in fluorescence indicates an increase in ROS levels. Results are representative of at least three independent replicates (*n* = 3). A *p*-value < 0.05 was considered statistically significant (represented by *) in addition to a *p*-value < 0.01 (represented by **), and a *p*-value < 0.001 (represented by ***).

**Figure 3 cancers-14-05478-f003:**
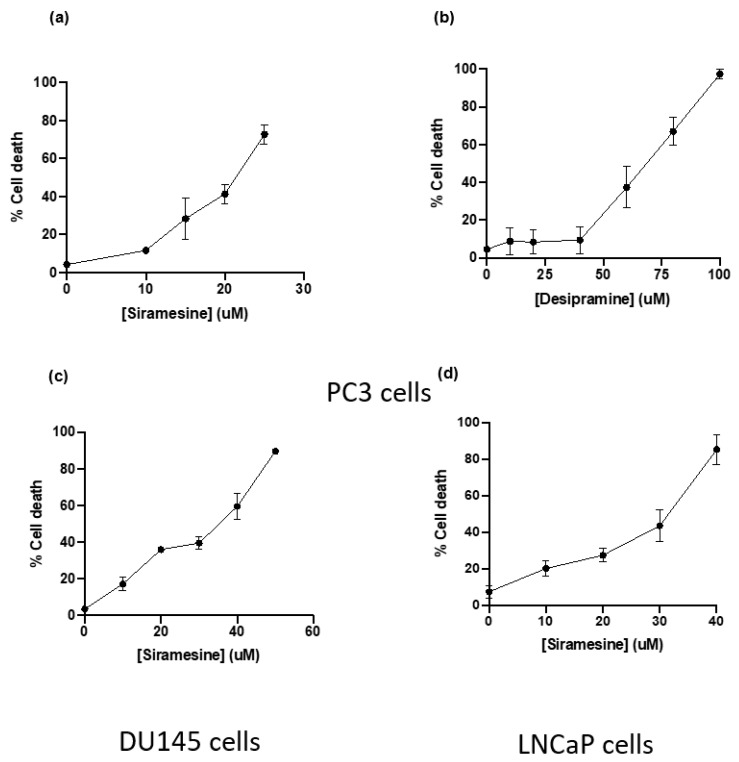
**Siramesine and desipramine induced cell death after 24 h treatment.** PC3 cells were treated with (**a**) siramesine and (**b**) desipramine for 24 h and stained with the fluorescent dye Trypan blue (0.4%) and analyzed by flow cytometry. An increase in fluorescence indicates an increase in cell death. Results are representative of at least three independent replicates (*n* = 3). (**c**) DU145 and (**d**) LNCaP cells were treated with increasing doses of siramesine for 24 h and stained with the fluorescent dye Trypan blue (0.4%) and analyzed by flow cytometry. An increase in fluorescence indicates an increase in cell death. Results are representative of at least three independent replicates (*n* = 3).

**Figure 4 cancers-14-05478-f004:**
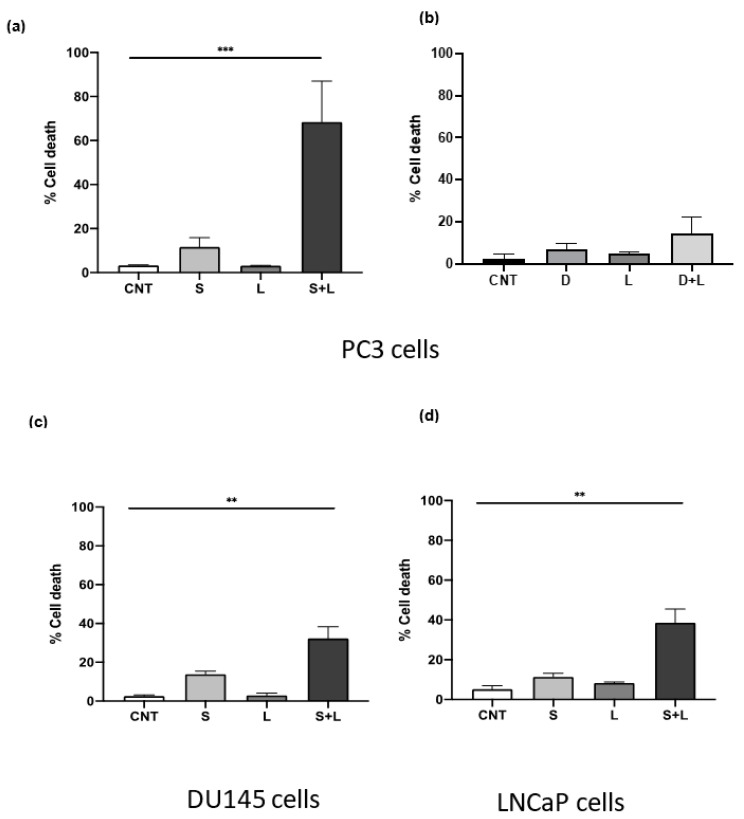
**Siramesine and lapatinib induce cell death in PC3 cells.** PC3 cells were treated with (**a**) siramesine (S, 10 μM) and lapatinib (L, 0.5 μM) or desipramine (D, 10 μM) and (**b**) Lapatinib (L, 0.5 μM) for 24 h and stained with the fluorescent dye Trypan blue (0.4%) and analyzed by flow cytometry. (**c**) DU145 cells or (**d**) LNCaP cells were treated as above 24 h. An increase in fluorescence indicates an increase in cell death. Results are representative of at least three independent replicates (*n* = 3). ** Represents statistical significant differences between control and combinational treatment (S + L) with a value of *p* < 0.05. For all other treatments there was no significant difference.

**Figure 5 cancers-14-05478-f005:**
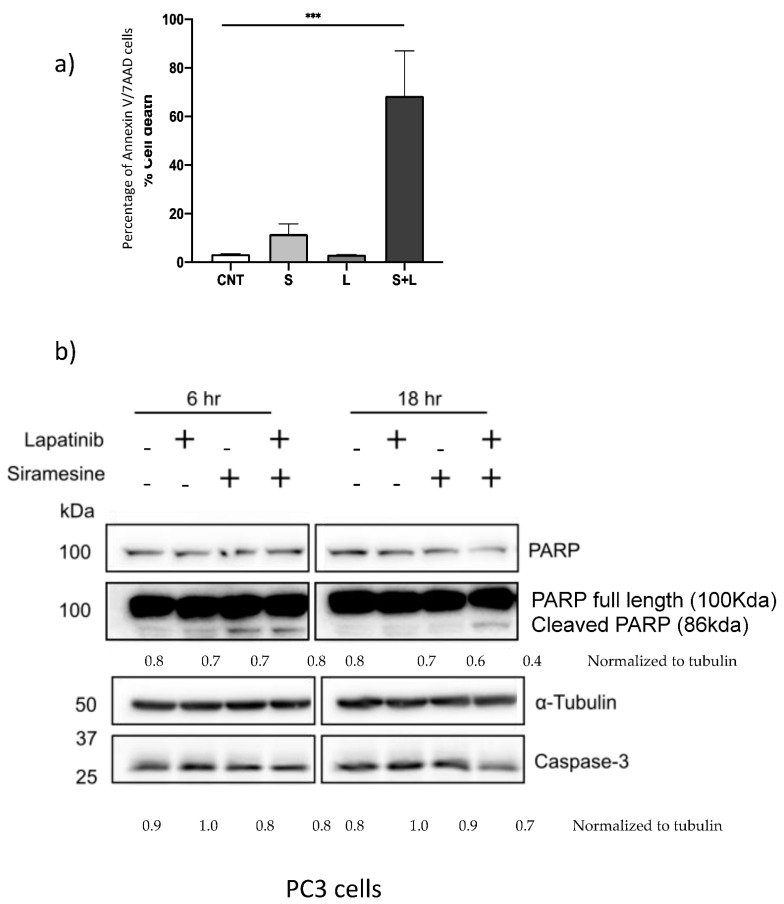
**Siramesine and lapatinib induce apoptotic cell death**. PC3 cells were treated with siramesine alone (S, 10 μM), lapatinib alone (L, 0.5 μM) or in combination. (**a**) After 24 h, cells were incubated with AnnexinV/7AAD dyes for 15 min and analyzed by flow cytometry. (**b**) Cells were lysed after 6 and 12 h of treatment and protein expression for PARP, and cas-3 were determined by western blot. Tubulin was used as a loading control a. Results are representative of three independent replicates (*n* = 3).

**Figure 6 cancers-14-05478-f006:**
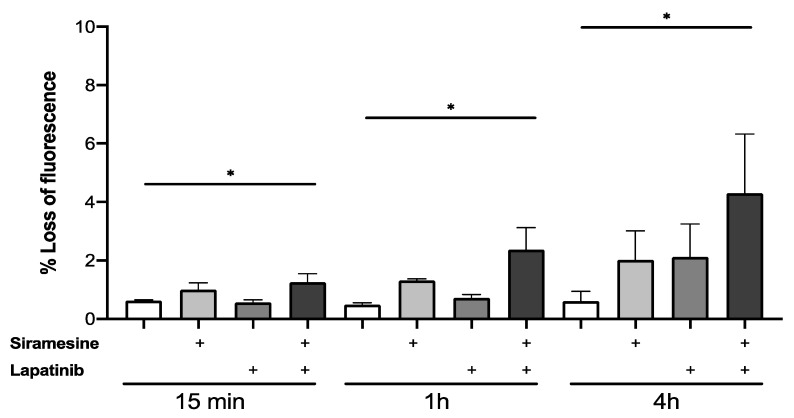
**Siramesine and lapatinib induce lysosomal membrane permeabilization in PC3 cells.** PC3 cells were treated with siramesine alone (10 μM), lapatinib alone (0.5 μM) or in combination for 15 min, 1 h and 4 h. Cells were stained with Lysotracker red (50 nM) for 30 min at 37 °C after treatment. LMP was quantified by flow cytometry where loss of fluorescence indicates lysosome membrane disruption. Results are representative of at least three independent replicates (*n* = 3). * represents a *p* value < 0.05.

**Figure 7 cancers-14-05478-f007:**
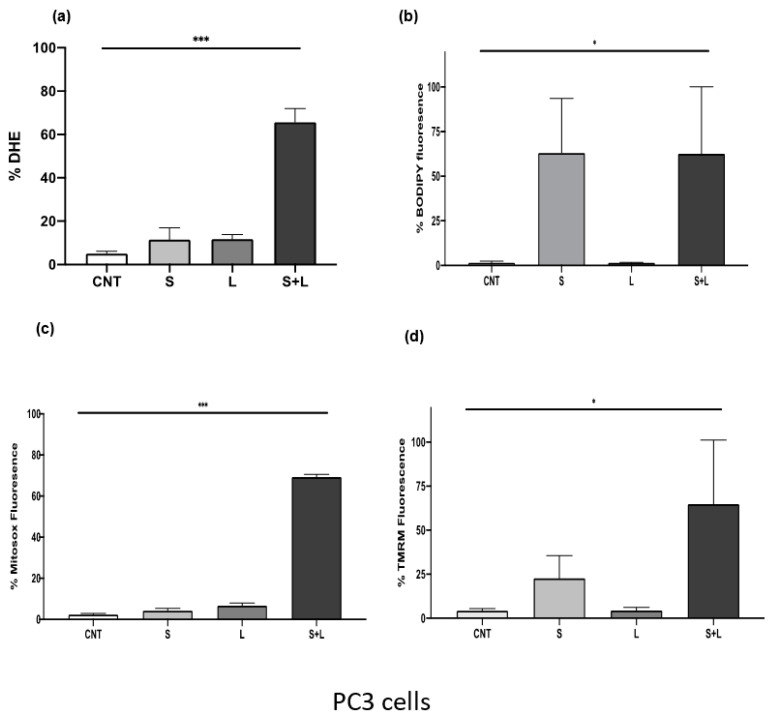
Siramesine and lapatinib increase reactive oxygen species, lipid peroxidation while decreasing mitochondrial membrane potential and generating mitochondrial super oxide. PC3 cells were treated with 10 μM siramesine and 0.5 μM lapatinib for 24 h. (**a**) To detect reactive oxygen species, cells were stained with 3.2 μM DHE for 30 min at 37 °C. An increase in fluorescence indicates an increase in reactive oxygen species. (**b**) Lipid peroxidation was detected by staining cells with 1 μM C11-BODIPY for 30 min at 37 °C. An increase in fluorescence indicates increased lipid peroxidation. for 30 min at 37 °C. An increase in fluorescence indicates an increase in reactive oxygen species. (**c**) Changes in mitochondrial membrane potential were measured by staining cells with 25 nM TMRM for 30 min at 37 °C. Increases in fluorescence indicate a decrease in mitochondrial membrane potential. (**d**) Mitochondrial superoxide levels were measured by staining cells with 5 μM Mitosox red for 10 min at 37 °C. An increase in fluorescence indicates an increase in mitochondrial super oxide. All these experiments were analyzed by flow cytometry. Results are representative of three independent replicates (*n* = 3). A *p*-value < 0.05 was considered statistically significant (represented by *) in addition to a *p*-value < 0.01 (represented by **), and a *p*-value < 0.001 (represented by ***).

**Figure 8 cancers-14-05478-f008:**
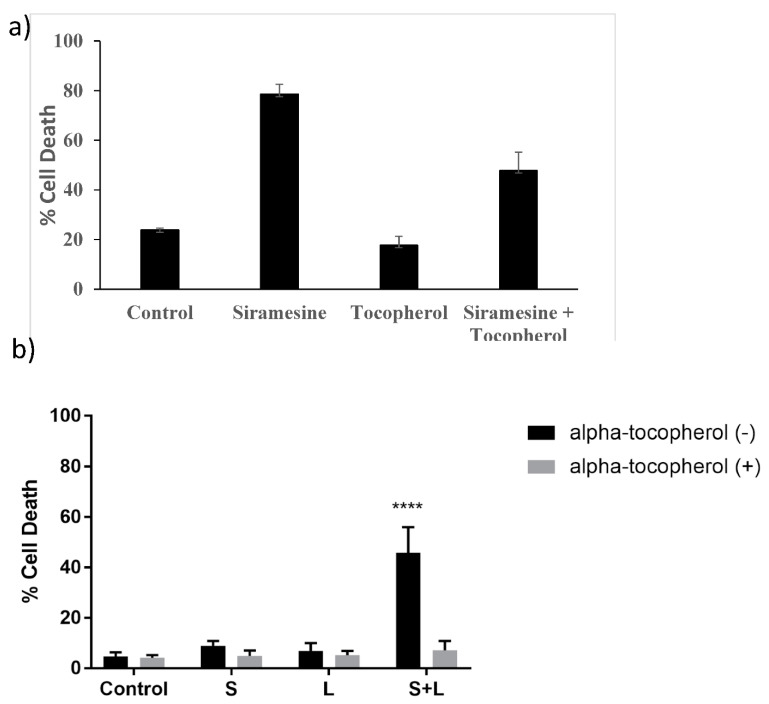
**Siramesine and lapatinib-induced cell death is blocked by antioxidant alpha-tocopherol**. (**a**) PC3 cells were treated with 10 μM siramesine for 24 h in the presence or absence of 10 μM alpha-tocopherol. Cell death was determined by Trypan blue exclusion assay. (**b**) PC3 cells were treated with 2.5 μM siramesine, 5 μM lapatinib or in combination in the presence or absence of alpha-tocopherol for 24 h. Cell death was determined by Trypan blue exclusion assay. The error bars represent standard error of three independent experiments and * presents statistical significance of *p* < 0.05.

**Figure 9 cancers-14-05478-f009:**
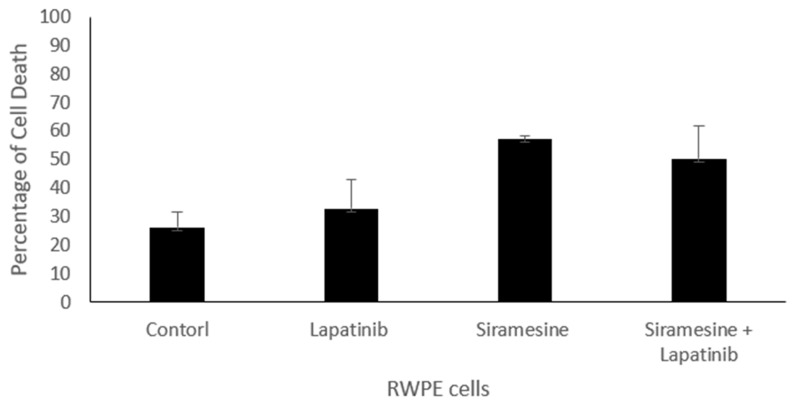
Siramesine and lapatinib treatment in non-transformed prostate epithelial cells RWPE. RWPE cells were treated with siramesine (10 mM) and/or lapatinib (0.5 mM) for 24 h. Control represent cells treat with DMSO. Amount of cell death was determined by Trypan blue exclusion assay. The error bars represent standard error and is the determine from three independent experiments.

## Data Availability

The data presented in this study are available upon request from corresponding author. The lysosomotropic agent siramesine and the tyrosine kinase inhibitor lapatinib induce cell death in prostate cancer cell.

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
