# Peer review of "Prostate Cancer Cells Are Sensitive to Lysosomotropic Agent Siramesine through Generation Reactive Oxygen Species and in Combination with Tyrosine Kinase Inhibitors"

_cancers, 2022, doi:10.3390/cancers14225478_

Round 1
Reviewer 1 Report (Previous Reviewer 3)
I have no further comments.
Author Response
Thank you for your acceptance of this manuscript.
Reviewer 2 Report (Previous Reviewer 5)
I recommend acceptance in present form.
Author Response
Thank you for your acceptance of this manuscript.
Reviewer 3 Report (New Reviewer)
The authors properly addressed the reviewers' major concerns but still there are some missing points.
One remaining concern about the PARP1 level shown in western blotting. The authors should clearly indicate whether the band was full-length (100-110 kD) or cleaved form (~86 kD). A wider range of bands showing both forms will provide better idea of whether PARP1 cleavage correlates with the observed increase in apoptosis.
Following this concern, the authors should put detailed information (catalog number, dilution) of the antibodies in the methods.
There are also remaining formating errors and typos. To name a few:
Several "LnCaP" are not corrected into "LNCaP". e.g. Figure 4. Please go through the manuscript carefully.
Line 266, PAPR should be PARP or PARP1.
The authors need to carefully update their manuscript again before acceptance.
Author Response
We will like to thank the reviewers for their kind comments on the manuscript. We have made the minor corrections requested. Please see below for specific reply to the comments.
One remaining concern about the PARP1 level shown in western blotting. The authors should clearly indicate whether the band was full-length (100-110 kD) or cleaved form (~86 kD). A wider range of bands showing both forms will provide better idea of whether PARP1 cleavage correlates with the observed increase in apoptosis.
Reply: We now indicate the full length PARP and the cleaved PARP in the western blot. This indicates caspase cleavage of PARP but at a lower level than the amount of apoptosis observed.
Following this concern, the authors should put detailed information (catalog number, dilution) of the antibodies in the methods.
Reply: The information requested is now indicated in the Material and Methods section.
There are also remaining formating errors and typos. To name a few:
Several "LnCaP" are not corrected into "LNCaP". e.g. Figure 4. Please go through the manuscript carefully.
Reply: This has been correct.
Line 266, PAPR should be PARP or PARP1.
Reply: The has been correct.
This manuscript is a resubmission of an earlier submission. The following is a list of the peer review reports and author responses from that submission.
Round 1
Reviewer 1 Report
Garcia et colleagues presents a nice work describing potential lysosomotropic agents with antineoplastic effect in prostate cancer. Although the topic is very interesting and relevant to the field, more experimental evidences are needed to support the conclusions of the manuscript.
- The base of the paper is the induction of lysosome membrane permeabilization (LMP) induced by certain compounds, like siramesine. However, LMP is measured based on the detection of lysotracker staining and no further validation is provided. Lysotracker is related to the lysosomal mass. Lysotracker staining is decreased if the lysosomal mass is decreased, but this phenomenon could be due to cytotoxicity (less cells), for example. There are several methods to prove that the lysosomal membrane is permeabilized. Additionally, microscopy images to check how the lysosomes are in treated cells, as compared to vehicle treated might be helpful.
- Cytotoxicity is shown as number of cells negative for trypan blue. It will be interesting to show cell viability using a more accurate method.
- In general, concentrations used in this paper are very high. Is there any rational to use this high micromolar concentrations?
- Annexin-V analysis is done by flow cytometry, but only a column graph is provided. It will be very interesting to provide the histogram plot.
- Synergism is not mathematically proven. There are a few methods that could be used here.
- Figures 1E and 1D seems to be mislabelled.
- Stats should be provided in Figure 4, for all the experimental conditions.
- TheWB in Figure 5B is not conclusive.
Author Response
There are several methods to prove that the lysosomal membrane is permeabilized. Additionally, microscopy images to check how the lysosomes are in treated cells, as compared to vehicle treated might be helpful.
Response: Thank you for this suggestion. We have in the past in our previous published work, show images of lysotracker in cells to determine lysosome disruption. These images do correlate with flow cytometry with lysotracker we presented in the manuscript. We find using flow cytometry with lysotracker more quantitative and reproducible whereas images using a fluorescent microscope only shows several cells in one field and is not quantitative. A presentative figure of our lysotracker staining was included in supplementary figure 1.
Cytotoxicity is shown as number of cells negative for trypan blue. It will be interesting to show cell viability using a more accurate method.
Response, Cell viability assay are a useful tool to determine the effects of cytotoxic agents on cell death and proliferation. In this case, we did not choose to use cell viability assay such as MTT assay as we have correlated our results with Annexin V apoptotic assay and feel it would not add new data.
In general, concentrations used in this paper are very high. Is there any rational to use this high micromolar concentrations?
Response: The concentrations were chosen for detection of cell death within 24 hours after treatment in vitro. This allows for mechanistic studies on the effects of this cells but will not reflect the clinically achievable doses for these drugs. To better determine the concentrations that will be effective in cancer patients, we will need to conduct animal studies in future directions and cell viability assays over 72 hour time courses. This going to be the focus of future investigation and is now reflected in the discussion of the manuscript.
- Annexin-V analysis is done by flow cytometry, but only a column graph is provided. It will be very interesting to provide the histogram plot.
Response: We have now added the Annexin V histogram to supplementary figure 4.
- Synergism is not mathematically proven. There are a few methods that could be used here.
Response: This is correct. We have conducted combinational index analysis with siramesine in previous papers (Chanas-Larue et al Leuekmia Research 2020, and Kost et al 2019 Cancers). For the sake of clarity, we have removed synergistic cell death and stated the combination increased cell death. The degree of synergism will be the focus of future investigations as we determine the achieve clinical doses for combinational treatments. This is now stated in the discussion.
- Figures 1E and 1D seems to be mislabelled.
Response: This has been corrected.
- Stats should be provided in Figure 4, for all the experimental conditions.
Response: This has been corrected.
- The WB in Figure 5B is not conclusive.
Response: We have added densitometry for this figure. We believe this figure by itself is not conclusive but in combination with caspase inhibitor failing to decrease combination induced cell death, we believe it provides an important contribution.
Reviewer 2 Report
The manuscript titled “Prostate Cancer cells sensitive to…..Kinase inhibitors,” by Garcia et al. demonstrates the synergistic activity of lysosomotropic agent siramesine and TK inhibitor
lapatinib in multiple prostate cancer programmed cell death. Although the data is exclusively cell-based observations, it is reliable and convincing, the findings are novel, and the conclusions are realistic. Scientific, statistical, and bioinformatics methods/analyses are fine. There are some major concerns.
1. It is not clear why PC3 cells were treated with Siramesine, Loratidine, Desloratadine, Clemastine, and Desiperamine, but only Siramesine treatment to DU 145 and LNCaP cells?
2. Figure, 1 and its legend have inaccurate marking e.g. The Figure has a to g, but the figure legend shows a and b. It shows that the authors are not serious about their experiments' presentation to the peer review.
3. Same comments (at in 2 above) for the experiments shown in figure 2.
4. The presented research work is only “observational,” and an obvious mechanistic aspect is lacking.
5. Since the siramesine is a lysosomotropic agent, it would make sense to demonstrate the autophagic aspect of the study. For example, is the cells 100% dying by the programmed cell death or Autophagy is also contributing to the destruction of the cancer cells?
6. Please be consistent with LNCaP as suppose to LnCap!
7. It is interesting to observe a lack of MCl-1 expression in Siramesine +Laptinib treat PC cells. Is it the same situation in DU145 and LNCaP cells too? This is interesting concerning post-transcriptional regulation events. If the protein biosynthesis is totally curtailed for the MCl-1, one should do an RT-qPCR to determine the levels of mRNA in treated cells.
8. It will be prudent to quantify the bands in all WB experiments by densitometric scans and plot a histogram for quantitative interpretation.
9. Why does PC 3 cell has a better response to siramesine + Laptinib treatments and not DU145 and LNCaP cells?
10. Since PC3 and DU145 lack androgen receptors and are castration-resistance cells, and LNCaP expresses AR and is castration-sensitive, the authors need to shed some light on the androgen-signaling aspect in their discussion.
Author Response
- It is not clear why PC3 cells were treated with Siramesine, Loratidine, Desloratadine, Clemastine, and Desiperamine, but only Siramesine treatment to DU 145 and LNCaP cells?
Response: We choose to treat PC3 cells as they are p53 mutated and resistant to most therapies compared to the other cell lines. This will identify the best lysosome disruptor to further test on other prostate cancer cell lines. We will elaborate in the results section.
- Figure, 1 and its legend have inaccurate marking e.g. The Figure has a to g, but the figure legend shows a and b. It shows that the authors are not serious about their experiments' presentation to the peer review.
Response: We apologize for the mistake, and it has been corrected.
- Same comments (at in 2 above) for the experiments shown in figure 2.
Response: We apologize again, and the figure has been corrected.
- The presented research work is only “observational,” and an obvious mechanistic aspect is lacking.
Response: As many of the experiments are observational, there are some result the lead to mechanistic aspects of the research such as the role of lipid peroxidation in the ability of siramesine to induce cell death. The reduction in Mcl1 protein levels could explain mitochondrial dysfunction. Further mechanistic studies will be the focus for future investigations and was added to the discussion.
- Since the siramesine is a lysosomotropic agent, it would make sense to demonstrate the autophagic aspect of the study. For example, is the cells 100% dying by the programmed cell death or Autophagy is also contributing to the destruction of the cancer cells?
Response: We have demonstrated that inhibiting autophagy fails to prevent siramesine induced cell death or in combination with lapatinib in prostate cancer cells (Supplemental figure 2). In addition, we failed to see autophagy changes in prostate cancer cells following treatment (data not shown). Thus, we do not believe autophagy is playing a major role in siramesine induced cell death in prostate cancer cells.
- Please be consistent with LNCaP as suppose to LnCap!
Response: We corrected the LNCaP spelling throughout the manuscript.
- It is interesting to observe a lack of MCl-1 expression in Siramesine +Laptinib treat PC cells. Is it the same situation in DU145 and LNCaP cells too? This is interesting concerning post-transcriptional regulation events. If the protein biosynthesis is totally curtailed for the MCl-1, one should do an RT-qPCR to determine the levels of mRNA in treated cells.
Response: The lack of Mcl-1 expression is interesting, and we have not yet determined whether it occurs in DU145 and LNCaP cells. Similarly, we have not determined whether the decrease is due to changes in transcription or degradation. This is one of the reasons it is in supplementary data as it corresponds to decreases in Mcl-1 in siramesine and lapatinib treatment in breast cancer cells previously published (Ma et al 2016 Cell death and Disease 7:e2307). Mcl-1 expression regulation will be the focus for future investigations, and we now discuss it in the manuscript.
- It will be prudent to quantify the bands in all WB experiments by densitometric scans and plot a histogram for quantitative interpretation.
Response: We have now included densitometric graphs for western blots (Figure 5 and supplemental figure 5) and histogram for annexin V/7AAD (Supplemental figure 4) .
- Why does PC 3 cell has a better response to siramesine + Laptinib treatments and not DU145 and LNCaP cells?
Response: We were surprised the PC3 cells had better responses to siraminesine + lapatinib treatments than DU145 or LNCaP cells. PC3 cells general are more resistant to cancer treatments due to p53 mutation. One potential explanation is due to the p53 loss, the metabolic pathways are altered leading to increased lysosome biogenesis. This would be a focus for future investigations
- Since PC3 and DU145 lack androgen receptors and are castration-resistance cells, and LNCaP expresses AR and is castration-sensitive, the authors need to shed some light on the androgen-signaling aspect in their discussion.
Response: We now have revised the discussion to discuss androgen-signaling in the context to lysosomes and siramesine sensitivity.
Reviewer 3 Report
This study was reported the utility of siramesine for prostate cancer cell. Generally, this paper is well written. The reviewer thinks that this paper has useful information for readers. However, the reviewer would like to suggest some critiques to make this paper as follows.
Major revision
1. On line 46, what is “affecting men”? The authors should revise this point.
2. Likewise, “Despite tremendous advancements….the quality of life of patients” is unclear. The authors should revise these sentences clearly.
3. On 56, “[6]but are” is wrong.
4. On line 60, what is ROS? The authors should spell out “reactive oxygen specirs.”
Author Response
- On line 46, what is “affecting men”? The authors should revise this point.
Response: We have revised the affecting men to prostate is a male form of cancer.
- Likewise, “Despite tremendous advancements….the quality of life of patients” is unclear. The authors should revise these sentences clearly.
Response: We have revised the quality of life of patients to be more clear.
- On 56, “[6]but are” is wrong.
Response: We have now corrected the mistake.
- On line 60, what is ROS? The authors should spell out “reactive oxygen specirs.”
Response: We have now corrected the mistake.
Reviewer 4 Report
Garcia et al. investigated in their manuscript "Prostate cancer cells are sensitive to lysosomotropic agent siramesine through generation reactive oxygen species and synergize to tyrosine kinase inhibitors" the effects of lysosomotropic agent and tyrosine kinase inhibitors on PCa. In general, the experiments are reasonably done, and the statistics seem to be in order, but the presentation of the Figures is sloppy and needs to be revised. Furthermore, the following items need to be modified before publication can be considered:
1. The manuscript investigates the effects of several mechanisms leading to a decrease in tumour mass. However, no cell viability or proliferation has been performed. Therefore, these experiments have to be added to the manuscript.
2. Drug concentrations vary between the different lysosomotropic agents, and therefore the comparison is not possible. Therefore, dose-response curves over a wide concentration range (at least 3 log steps) must be performed to make a true statement about effectiveness.
3. The authors claim a synergistic effect of their drug combination. However, this has not been mathematically proven. I highly recommend the Tallarida "Quantitative Methods for Assessing Drug Synergism" review and performing an isobologram analysis. At the moment, the statement "synergistic effects" is just a lucky guess
4. Densitometry of their western blot, especially cPARP and caspase 3, must be performed. Moreover, there is no explanation why Apoptosis analysis by flow cytometry has been completed after 24 h and western blot after 18 h
5. The experiments with normal cells (RWPE) should be shown in the main figures as it is essential to show cancer-specific effects.
6. Line 336: To this reviewer's knowledge, paclitaxel and etoposide are not a state of the art treatments. Please add the guidelines recommending these drugs.
7. Line 352: "Data not shown" cannot be accepted anymore. Please provide all data.
Author Response
- The manuscript investigates the effects of several mechanisms leading to a decrease in tumour mass. However, no cell viability or proliferation has been performed. Therefore, these experiments have to be added to the manuscript.
Response: We have investigated cell death using two different methods (membrane permeabilization and Annexin V staining). Cell viability assays we have done in the past and showed similar results to cell death assays (Dielschneider et al 2016 Leukema, 30:1290, Ma et al 2016 Cell Death and Disease 7:e2307). One issue with cell viability assays is they measure both cell death and proliferation. Effects on proliferation was not the focus of this study.
- Drug concentrations vary between the different lysosomotropic agents, and therefore the comparison is not possible. Therefore, dose-response curves over a wide concentration range (at least 3 log steps) must be performed to make a true statement about effectiveness.
Response: We used a range of drug concentrations on PC3 cells to determine the most cell death after 24 hours for a lysosomotropic agent to use on subsequent experiments. We decided not to use a wide range of concentrations (3 log steps) as this will not be clinically achievable.
- The authors claim a synergistic effect of their drug combination. However, this has not been mathematically proven. I highly recommend the Tallarida "Quantitative Methods for Assessing Drug Synergism" review and performing an isobologram analysis. At the moment, the statement "synergistic effects" is just a lucky guess
Response: This is a good point and we have removed the word synergy from our description of the combination of siramesine and lapatinib treatments.
- Densitometry of their western blot, especially cPARP and caspase 3, must be performed. Moreover, there is no explanation why Apoptosis analysis by flow cytometry has been completed after 24 h and western blot after 18 h
Response: We have now included densitometry for PARP and caspase 3. The reason the western blots were done earlier times than apoptosis is changes in proteins become harder to determine as cells degrade into apoptotic bodies. It biases the experiment for cell that survive after treatment. Thus, we need earlier times to detect changes in protein levels during cell death.
- The experiments with normal cells (RWPE) should be shown in the main figures as it is essential to show cancer-specific effects.
Response. The figure is now part of the main manuscript (Figure 9)
- Line 336: To this reviewer's knowledge, paclitaxel and etoposide are not a state of the art treatments. Please add the guidelines recommending these drugs.
Response: We added references to the use of these drugs in prostate cancer treatment and altered the statement to indicate these drugs have been used to treat prostate cancer.
- Line 352: "Data not shown" cannot be accepted anymore. Please provide all data.
Response: These statements have now been removed.
Reviewer 5 Report
I consider the present manuscript entitled “Prostate cancer cells are sensitive to lysosomotropic agent siramesine through generation reactive oxygen species and synergize to tyrosine kinase inhibitors” having an interesting topic, with a clear and easy title to be individualized on literature research title. The abstract is well structured, but it is not clearly stated the aim of the study.
The Introduction is too short, with few data about prostate cancer. I recommend a larger description of metabolic lysosomal and tyrosine-kinase processes in prostate cancer. The objectives of the study should be included in the last paragraph of Introduction Section.
Material and Methods are well conceived, to make the study reproducible.
The Results cover all the required fields. I recommend a larger description of lysosomotropic agents mechanism of action in cells culture in Material and Methods – Drug treatments, not in Results section.
Discussions are well conceived and support the Results. In this section no study limitations and the impact of the study on the literature research are presented. The Conclusions reflect the idea of the title and the manuscript presents a recent bibliography, but with a small number of titles.
Author Response
The Results cover all the required fields. I recommend a larger description of lysosomotropic agents mechanism of action in cells culture in Material and Methods – Drug treatments, not in Results section.
Response: We have increased the description of lysosomotropic drugs in the material and methods section.
In this section no study limitations are presented.
Response: We now have expanded the discussion to include study limitations for cell lines used and clinically achievable doses. Future studies will be conducted to address these limitations.
Round 2
Reviewer 1 Report
As stated before, lysotracker is not a marker of LMP. LMP can be detected by Galectina puncta, cathepsin release, among others techniques. As explained before, lysotracker measures the lysosomal mass, not the permeabilization of membrane. As LMP is the basis of this paper, unless clearly demonstrated, the importance of the results are limited.
Unfortunatelly, the authors have not even tried to discuss the fact that trypan blue staining is used for cell viability quantification, when there are alternative methods more acqurated and widely used.
The fact that there is no synergism reduces the importance/impact of the findings and the full discussion should be reorientated.
The concentrations used in this manuscript are several magnitudes above the highest recommended for preclinical work. Again, the importance of the current manuscript in biomedicine is limited.
Author Response
As stated before, lysotracker is not a marker of LMP. LMP can be detected by Galectina puncta, cathepsin release, among others techniques. As explained before, lysotracker measures the lysosomal mass, not the permeabilization of membrane. As LMP is the basis of this paper, unless clearly demonstrated, the importance of the results are limited.
Response: We agree lysotracker is not a direct assay to access LMP as it measures loss of acidic vesicles. The loss of acidic vesicles could be explained either by LMP or by an increase in lysosomal pH independent of LMP. Previous studies have shown siramesine induing LMP using other techniques such as galectin assay (Autophagy. 2015 Aug; 11(8): 1408–1424). We do acknowledge that flow cytometry fails to consider puncta lysotracker staining. We stain PC3 cells with lysotracker and visualize the cells through fluorescent microscopy. We found punctate staining for lysotracker which is reduced by siramesine and lapatinib treatment (Supplemental figure 6).
Unfortunately, the authors have not even tried to discuss the fact that trypan blue staining is used for cell viability quantification, when there are alternative methods more acqurated and widely used.
Response: We have now confirmed the results with trypan blue exclusion with a cell viability assay (MTS) presented in Supplemental Figure 1.
The fact that there is no synergism reduces the importance/impact of the findings and the full discussion should be reorientated.
Response: We have conducted a combination index (CI) analysis and found the combination with siramesine and lapatinib gave CI=0.7 indicating synergism.
The concentrations used in this manuscript are several magnitudes above the highest recommended for preclinical work. Again, the importance of the current manuscript in biomedicine is limited.
Response: We calculated the IC50 for lapatinib (2.75mM) and siramesine (19mM) in PC3 cells. The maximal plasma concentration for lapaitinib is 4.18mM (CCR 2017 23:3489) and siramesine has not been clinically tested for cancer treatment so clinically achievable dose are unknown. To determine clinical dose, we will need to test this treatment in animal models which will be the focus of future studies.
Reviewer 4 Report
In the new version of the manuscript, the authors discussed the main issues mentioned by this reviewer. However, their modifications do not justify acceptance as the main problems just have been deleted or ignored. The following main issues are still not solved:
1. The manuscript has no data on cell viability, only short-term apoptosis data. As the authors focus on molecular mechanisms and not cell growth data, a more detailed analysis of apoptosis pathways should be included. Moreover, as the authors try to sell their combination as a new therapeutic strategy, proliferation data cannot be avoided. Moreover, clonogenic recovery assays should be performed to assess the recovery potential after treatment.
2. Dose-response curves and calculations of IC50 are still necessary to compare the drugs. µM concentrations are in general high and may not be clinically achievable.
3. Just deleting “synergistic” is not the solution to the issue. Combining therapies should always address whether the effects are synergistic or just additive.
4. The added references for the use of paclitaxel and etoposide are not a state of the art treatments and are not valid. Please add clinical guidelines or correct the issue.
5. Deleting “Data not shown” without adding the data is highly suspicious.
6. Minor comment: Changing all LnCaP to LNCaP should also be performed in the figures.
Author Response
- The manuscript has no data on cell viability, only short-term apoptosis data. As the authors focus on molecular mechanisms and not cell growth data, a more detailed analysis of apoptosis pathways should be included. Moreover, as the authors try to sell their combination as a new therapeutic strategy, proliferation data cannot be avoided. Moreover, clonogenic recovery assays should be performed to assess the recovery potential after treatment.
Response: We have now conduct cell viability assay (MTS) on PC3 cells and showed similar cell death response to siramesine and lapatinib to trypan blue exclusion (Supplemental Figure 1).
- Dose-response curves and calculations of IC50 are still necessary to compare the drugs. µM concentrations are in general high and may not be clinically achievable.
Response: We calculated the IC50 for lapatinib (2.75mM) and siramesine (19mM) in PC3 cells. The maximal plasma concentration for lapaitinib is 4.18mM (CCR 2017 23:3489) and siramesine has not been clinically tested for cancer treatment thus clinically acheivalbe doses are unknown. To determine clinical dose, we will need to test this treatment in animal models which will be the focus of future studies.
- Just deleting “synergistic” is not the solution to the issue. Combining therapies should always address whether the effects are synergistic or just additive.
Response: We have conducted a combination index (CI) analysis and found the combination with siramesine and lapatinib gave CI=0.7 indicating synergism. This is now include in the manuscript.
- The added references for the use of paclitaxel and etoposide are not a state of the art treatments and are not valid. Please add clinical guidelines or correct the issue.
Response: It was not our intension to have state of the art treatments for these experiments. It was to show chemotherapy treatments fail to give similar responses. I have changed wording to correct this issue.
- Minor comment: Changing all LnCaP to LNCaP should also be performed in the figures.
Response: Changed in figures.